# Germplasm Enhancement and Identification of Loci Conferring Resistance against *Plasmodiophora brassicae* in Broccoli

**DOI:** 10.3390/genes13091600

**Published:** 2022-09-07

**Authors:** Qi Xie, Xiaochun Wei, Yumei Liu, Fengqing Han, Zhansheng Li

**Affiliations:** 1Key Laboratory of Biology and Genetic Improvement of Horticultural Crops, Institute of Vegetables and Flowers, Chinese Academy of Agricultural Sciences, Beijing 100081, China; 2Institute of Horticulture, Henan Academy of Agricultural Sciences, Zhengzhou 450002, China

**Keywords:** *Plasmodiophora brassicae*, clubroot, broccoli, germplasm enhancement, breeding

## Abstract

In order to breed broccoli and other *Brassica* materials to be highly resistant to clubroot disease, 41 Brassicaceae varieties were developed and identified between 2020 and 2021. Seven known clubroot genes were used for screening these materials. In addition, the resistant and susceptible broccoli cultivars were designed for observing their differences in the infection process with *Plasmodiophora brassicae*. The results showed that 90% of total materials had carried more than two clubroot resistance genes: one material carried two disease resistance genes, four materials carried seven genes for clubroot resistance, two materials carried six genes for clubroot resistance, and in total 32% of these materials carried five genes for clubroot resistance. As a result, several new genotypes of Brassicaceae germplasm were firstly created and obtained based on distant hybridization and identification of loci conferring resistance against *Plasmodiophora brassicae* in this study. We found and revealed that similar infection models of *Plasmodiophora brassicae* occurred in susceptible and resistant cultivars of broccoli, but differences in infection efficiency of *Plasmodiophora brassicae* also existed in both materials. For resistant broccoli plants, a small number of conidia formed in the root hair, and only a few spores could enter the cortex without forming sporangia while sporangia could form in susceptible plants. Our study could provide critical *Brassica* materials for breeding resistant varieties and new insight into understanding the mechanism of plant resistance.

## 1. Introduction

Clubroot, a major disease of Brassicaceae crops caused by *Plasmodiophora brassicae* (*P. brassicae*), is an economically important soil borne disease worldwide. It was first discovered in the Mediterranean and southern Europe but now occurs all over the world and has become one of the most serious diseases of cruciferous crops including cauliflower, cabbage, broccoli, Chinese cabbage, turnip, radish and rapeseed [1,2,3]. Due to the long survival time of its spores in soil, traditional agricultural control methods usually have little effect. Therefore, the effective ways to control the spores are germplasm enhancement and breeding of resistant varieties of *Brassica* plants [4,5]. Therefore, creating Brassicaceae germplasms resistant to clubroot for crop breeding was a fundamental method and essential to crop production.

Broccoli (*Brassica oleracea* L. var. *italica*) is an important member of the Brassicaceae family and widely cultivated as a popular vegetable crop worldwide. It is rich in vitamins, proteins and minerals, as well as some anticancer bioactive compounds, such as sulforaphane and indole-3-carbinol [6,7,8]. More than 80,000 ha of broccoli has been cultivated in China, which has become the largest producer of broccoli in the world [9]. In recent years, clubroot disease of broccoli has been noted in Zhejiang, Henan, Yunnan and Shandong provinces, which were the dominant broccoli planting areas in China. Therefore, it was necessary to create broccoli resistance materials for future breeding [10,11].

To date, most studies have focused on the location and mining of resistance loci and the verification of their function. So far, four clubroot resistance (CR) genes have been cloned, including *Crr1*, *CRa*, *CRd* and *CRb^kato^*. *Crr1* and *CRa* belong to the TIR-NB-LRR gene family and are found in mainly turnip and Chinese cabbage [1,12,13,14]. Due to the diversity of physiological strains and the strong strain specificity of single-gene disease-resistant materials, long-term planting of single-gene disease-resistant varieties will lead to a loss of resistance [13]. In the future, we will carry out multi-resistance site polymerization breeding and cultivate varieties with high resistance to multiple physiological strains of *P. brassicae* to address the continuous harm to cruciferous plant production.

There exist differences in diversity and pathogenicity among countries, regions and root disease seasons [15]. The purpose of the study was to screen clubroot resistant germplasms developed by distant hybridization from cruciferous materials in 2020 and 2021. These materials were collected based on previous reported crops (turnip, cabbage and oilseed rape). At the same time, some possible identifying molecular markers to rapidly detect CR genes among Brassicaceae resources were used for detecting all materials in the study. Therefore, our study would facilitate resource innovation and improvement of materials in the *Brassica* genus to develop varieties with multiple CR genes. The aim of this study was to provide essential materials and theoretical support for the breeding of new varieties of broccoli and the other *Brassica* crops with multiple CR genes, as well as to provide new insight into understanding the mechanism of clubroot infection in crops [16,17].

## 2. Materials and Methods

### 2.1. Plant Materials

Cruciferous resources from different regions of China, including the Yangtze River Basin and northwest, southeast and central regions, were collected and then improved. All the materials were preserved and planted at the experimental station of Institute of Vegetables and Flowers, Chinese Academy of Agricultural Sciences (IVF-CAAS). The genetic information of the 41 cruciferous accessions is shown in Table 1.

### 2.2. Molecular Markers and Identification of Clubroot Resistance Genes

Ten pairs of primers for clubroot resistance markers described in previous studies were used to identify CR genes in the experimental materials (Table 2). Among these primers, SC2930-T-FW/SC2930-RV and SC2930-Q-FW/SC2930-RV were linked to the clubroot resistance gene *CRa*, KBrH129J18R was linked to *CRb*, B50-C9-FW/-RV and B50-6R-FW/-RV were linked to *CRc*, HC688-4-FW/-7-RV was linked to *CRk* [18], OPC11-2S was linked to *Crr3* and BRMS-088 and BRMS-096 were linked to the genes of *Crr1* and *Crr2*, respectively [12,13,19].

Genomic DNA was extracted from young leaves using the improved cetyltrimethylammonium bromide (CTAB) method [7,9], and the reagents for PCR were purchased from Vazyme Biotech Co., Ltd. (Virginia, USA). The PCR amplification was carried out in a 10.0 µL reaction mixture containing 2.0 µL of DNA template, 2.0 µL of buffer, 1.0 µL of primers and 5.0 µL of Taq Plus Master Mix. PCR products larger than 300 bp were identified using 1% agarose gel electrophoresis followed by visualization of the gel in an automatic gel imaging system. PCR products shorter than 300 bp were developed using 8.0% polyacrylamide gel electrophoresis and silver staining.

### 2.3. Pathogen Inoculation and Molecular Verification

Root gall samples from Chinese cabbage plants were collected from Xinye, Henan province, and the pathogen was identified and reported as race 4 [20,21,22]. Inoculation and resistance testing were performed using our previously described method [23]. DNA was extracted from the gall samples using the CTAB method, and PCR amplification and identification were carried out with the previously reported clubroot strain-specific primers (Table 3). The PCR system was the same as that used for the identification of clubroot resistance loci. The amplification protocol was as follows: 94 °C for 3.0 min (predenaturation); denaturation at 94 °C for 30.0 s, annealing at 55 °C for 30.0 s and extension at 72 °C for 45.0 s; a final extension at 72 °C for 7.0 min; and storage at 10 °C. The amplification products were separated on a 1.0% agarose gel.

### 2.4. Staining and Observation

The susceptible broccoli cultivar B991 and the resistant cultivar “Yacui91” were selected for the hydroponic experiments. After germination, 24 seedlings of each variety were retained for observation. The infection process was carried out in a hydroponic solution system. When the broccoli seedlings produced one true leaf, they were moved to a centrifuge tube with the standard bacterial solution of 4 × 10^7^ spores/mL. The system was put in an artificial climate incubator under the following conditions: 16.0 h of light at 25 °C, 8.0 h of dark at 20 °C with the relative humidity of 75.0%. At 0th, 7th and 14th days, the samples of broccoli root hairs and cortex were gathered and treated for infection observation, 3 individual plant distribution observations were selected at each time point. Firstly, broccoli root was placed in formal-acetic-alcohol (FAA) fixative for 24.0 h, and then FAA solution fixative was washed off with distilled water, stained in 0.5% Phloxine for 20.0 min and rinsed well with water. Finally, we conducted microscopic observations. Three replicates were selected for observation and imaging (n = 3).

### 2.5. Ploidy Detection by Flow Cytometry

Leaves were sampled from plants with five or more disease resistance genes, and broccoli (CC) and oilseed rape (AACC) were used as controls to establish the population for measuring the DNA levels. The corresponding peak value of the G_0_/G_1_ phase was adjusted to 200. We treated a 200-mg sample with 2.0 mL of Galbraith’s buffer, and a disposable blade was used to chop the tissue. After filtering through a 400-mesh filter, the liquid was added to a 2 mL centrifuge tube and centrifuged at 500 r/min for 3.0 min, then, the supernatant was discarded. Finally, PI dye solution was added to stain the nuclei, and the cells were placed on ice for 30.0 min. The nuclei were visualized by a microscope [24,25,26].

## 3. Results

### 3.1. Diversity of Clubroot Resistance and Plant Ploidy

The results showed that seven clubroot resistance genes, namely, *CRa*, *CRb*, *CRc*, *CRk*, *Crr1*, *Crr2* and *Crr3*, were all well amplified in the test materials (Table 4) (Figure 1). Among these CR genes, four materials were detected in 2020, and *CRa* was found only in black mustard B1086. The CR gene *Crr3* was detected in all the materials except Porphyra yezoensis and black mustard. The other materials contained homozygous *Crr2* genes. In 2021, some varieties were detected with seven CR genes. Only the homozygous and heterozygous CR gene of *CRk* were detected in oilseed rape and the hybris cross between cabbage and Chinese cabbage. In this study, *CRa* and *CRb* genes were widely detected in all the materials, but *CRa* was absent in two varieties of B571 and B582. Homozygous and heterozygous CR genes were all detected in the oilseed rape varieties “Huayouza62R” (B578), B1024, B1025 and B1026. The *CRc* gene was found in Chinese cabbage, oilseed rape and turnip. The heterozygous *Crr1* gene was detected in “Huayouza62R”, turnip, “Yacui91” and broccoli hybrids. It is noteworthy that “Yacui91” and European mustard contained only *CRb* gene. The CR gene of *Crr2* was present in all the test materials except European mustard, “Yacui91” and broccoli. B613 and B614 were hybrids cross between cabbage and turnip without *CRk* gene. Homozygous and heterozygous CR genes could be detected in the other six varieties. Chinese cabbage named B608 was obtained from the National Germplasm Bank and harbored four CR genes. In our study, most of the materials contained two or more CR genes. The hybrids crossed between Chinese cabbage and “Huayouza62R” contained seven CR genes, the hybrids crossed between cabbage and turnip contained six CR genes, and the other materials contained five CR genes. The growth diagram of materials and their aggregated resistance genes were shown in Figure 2.

As shown in Table 4 and Figure 2, morphological diversity and CR genes were found in these improved cruciferous germplasms. At the same time, the detection of various ploidy levels via DNA content measurement also verified the genetic information (Figure 3). As shown in Figure 3, there were six diploid materials in total, which were B606, B611, B831, B1019, B1024 and B1027. B606 and B611 were BC1F_3_ generations of red cabbage (*B. oleracea* L. var. *capitata*) × turnip (*B. rapa* L. ssp. *rapa*), B831 was the BC1 generation of broccoli (*B. oleracea* L. var. *italica*) × wild cabbage (*B. macrocarpa* Guss.) × rape (*B. napus* L.), B1024 was the F_1_ generation of choi sum (*B. campestris* L. ssp. *chinensis* var. *utilis Tsen et Lee*) × Chinese cabbage (*B. pekinensis Rupr*.) and B1027 was the BC1 generation of broccoli (*B. oleracea* L. var. *italica*) × rape (*B. napus* L.). B578 was a tetraploid oil rape variety. In addition, B368, B910 and B936 were identified as triploid, their fluorescence peaks were between 200 and 400, B910 and B936 belonged to F_1_ and BC1 was the generation of broccoli (*B. oleracea* L. var. *italica*) × turnip (*B. rapa* L. ssp. *rapa*).

### 3.2. Disease Identification and Phylogenetic Analysis of CR Genes

In this study, six pairs of specific primers, namely, novel342-2, novel407-2, PBRA 007750-2, PBRA_008439-1, PBRA_009348-1 and actin1, amplified the target bands (Figure 4a), which might infer that the main strain collected in Xinye was a pathogen 4 [3,23]. This result was consistent with previous reports and our identification. Moreover, “Huayouza62R” (B578), turnip (B581) and broccoli “Yacui91” exhibited high pathogen resistance (Figure 4c) (Table 1). The results of individual detection in hybrids of cabbage and turnip showed that 4 of 31 individual B612 plants were diseased, and 3 of 29 individual B613 plants were diseased, while only 1 of 58 individual B614 plants was diseased. The ratio of resistance to susceptibility was 59 to 4, and some traits were separated in the tested offspring, which was consistent with the results of molecular identification.

In addition, we evaluated the evolutionary relationships of CR genes using the maximum likelihood method in MEGA7 software (Auckland, New Zealand). As shown in Figure 4b, the CR genes could be divided into three branches with a genetic similarity coefficient of 1.1; there were three genes in subgroup I, namely, *CRa*, *CRb* and *Crr1* and *CRa* was the same as *CRb* reported in previous studies [1,3,18]. Subgroup II included only one CR gene, *Crr3*, suggesting that the relationship was different from the others. *Crr3* and *CRc* were clustered together and belonged to subgroup III.

### 3.3. Comparisons of the Infection Processes of P. brassicae in Resistant and Susceptible Broccoli Cultivars

The root hairs of broccoli infected by *P. brassicae* were red, and the uninfected root hairs were colorless. We found that the root sections were initially not infected as shown in Figure 5a. Moreover, a large number of zoospores were observed in the susceptible broccoli roots, indicating that *P. brassicae* could easily infect the root hairs. As shown in Figure 5, some resting spores entered the cortical cells from the root hairs, producing many zoospores (Figure 5b). *P. brassicae* obviously infected the root cortical cells, as shown in Figure 5c, and a large number of sporangia were formed in the susceptible broccoli variety B991, inducing cortical infection. With the extension of infection time, the root hair infection rate on the 14th day was significantly higher than that on the 7th day. *P. brassicae* was found to rapidly infect susceptible broccoli on the 7th and 14th days.

As shown in Figure 5e,f, a different result was obtained. *P. brassicae* began to infect the root hairs of the resistant broccoli variety “Yacui91” on the 7th day, but fewer root hairs were infected. Although *P. brassicae* could invade the root cells of broccoli “Yacui91” on the 14th day, fewer zoospores and secondary zoospores were found in the root cells, and no sporangium formation was observed. At the same time, the root hair infection rate on the 14th day was significantly lower than that on the 7th day in resistant broccoli than in susceptible broccoli.

## 4. Discussion

According to previous studies, these CR genes are initially absent in *B. oleracea* plants and are usually different from those in *Brassica pekinensis*. After extensive screening of *B. oleracea* species, we found that most varieties of broccoli, cabbage, cauliflower, Chinese kale and kohlrabi were susceptible, and only a few materials were resistant and potentially useful in breeding. We also clarified the relationship between the disease resistance loci and the characteristics of different materials in each cruciferous family [10,27]. In this study, a number of cruciferous species were created and identified by major possible CR genes. To date, it is reported that both *CRa* and *CRb* are located on chromosome A03, and *CRa* is located on chromosome A03. Moreover, two RFLP molecular markers linked to the anti-root-tumor gene were obtained [19]. *CRb* and its linked markers TCR05 and TCR09 in disease-resistant Chinese cabbage and turnip ECD01 have been identified and it is predicted that they may be alleles or two closely linked disease resistance loci [5,28,29]. It was reported that *CRb* and *CRa* were two closely linked disease resistance loci [30]. In our study, only *CRa*, not *CRb*, was detected in broccoli, cabbage, rape, black mustard and some turnip and Chinese cabbage hybrids, and the results for these two loci were not completely consistent among the different materials, which might indicate gene preferences of CR genes occurred in the *Brassica* species [31]. The *Crr3* gene has been fine-mapped through comparison with *Arabidopsis thaliana* [32], and the *CRk* gene was also found to be located on the Chinese cabbage A03 chromosome in 2008 [33]. It has been proposed that *Crr3* and *CRk* may also be alleles or two closely linked disease resistance loci [3]. However, in this experiment, *CRk* was the only detected homozygous-susceptible locus in turnip and black mustard, while *Crr3* was detected in all of these clubroot resistant materials. From this result, we inferred that *CRk* might be a more efficient major gene. It has been reported that *CRd* is located in a 60-kb region between chromosome markers yau389 and yau376 on chromosome A03 [3]. Based on the physical location of the *Crr3* linkage marker, *CRd* was also located upstream of *Crr3*; meanwhile *CRa*, *CRb* and *CRb^Kato^* were located in the region between 23.6 Mb and 25.6 Mb of the Chinese cabbage A03 linkage group [30] in which *CRb* was identified in the area of 23.6–23.7 Mb and *CRa* was located between 24.2 Mb and 24.5 Mb (Figure 6). Therefore, this study provides necessary CR materials to help us understand the efficiency of these reported genes in cruciferous plants.

European turnip and some oilseed rape varieties contained several important resistance genes that conferred vertical resistance to different physiological strains of *P. brassicae*. By constructing a genetic map of a disease-resistant segregated Chinese cabbage population derived from the Siloga turnip, three genes, namely, *Crr1*, *Crr2* and *Crr4*, were identified, and *Crr1* was found to be located on chromosome A08 [12]. The homologous sequence information of *Arabidopsis thaliana* was used for cloning, and *Crr2* was located on chromosome A01, while *Crr4* was located on chromosome A06. In our study, the experimental materials were all identified using published clubroot resistance markers, and it was found that most of the materials contained *Crr1*, *Crr2* and *Crr3* resistance sites. The *CRc* gene was detected in only four crops: broccoli and rape hybrids, rape, black mustard and cabbage and turnip hybrids. Ninety percent of the materials contained two or more disease resistance genes. The results of the molecular marker detection theoretically revealed the reasons for disease resistance.

In this a pioneering study, the infection process of *P. brassicae* in resistant and susceptible broccoli under hydroponic conditions was elucidated, and the result might provide a rapid and reliable detection method for clubroot compared to traditional methods usually requiring more than 30 days. Moreover, the differences in *P. brassicae* infection in resistant and susceptible broccoli suggested that, although the resting spores could be transformed to zoospores in root hairs of resistant broccoli plants, the sporangium could be formed, and the formation of more zoospores and its invasion were prevented after the 10th day in the hydroponic environment. However, in susceptible broccoli plants, *P. brassicae* could achieve rapid infection and produce a large number of conidia and sporangia. The reason for this large difference might be closely related to the composition of the root exudates, which needs to be further studied. Therefore, this study also provided new evidence and approaches for studying the molecular mechanisms of clubroot resistance in *Brassica* plants [34,35]. Finally, germplasm enhancement and identification of loci conferring resistance against domain pathotypes of *P. brassicae* in broccoli and other cruciferous species would be beneficial for breeding disease-resistant varieties [10,27,36].

## 5. Conclusions

Based on 41 Brassicaceae plants, we have firstly created and then obtained six essential materials together with six or more CR genes currently lacking in *Brassica* crops. Meanwhile, a total of 32% of all materials carried five CR genes, which is helpful for breeding disease-resistant varieties of broccoli and the other *Brassica* crops. Here we provide these essential materials which will be beneficial to more in-depth research in *Brassica* crops and international co-operation in the exchange of clubroot resistant materials for better global breeding of cruciferous crops. Furthermore, in this study, we also explored and discussed the profiles of *P. brassicae* infecting the resistant and susceptible materials of broccoli, which provide new insight into the infection mechanism of clubroot in *Brassica* crops.

## Figures and Tables

**Figure 1 genes-13-01600-f001:**
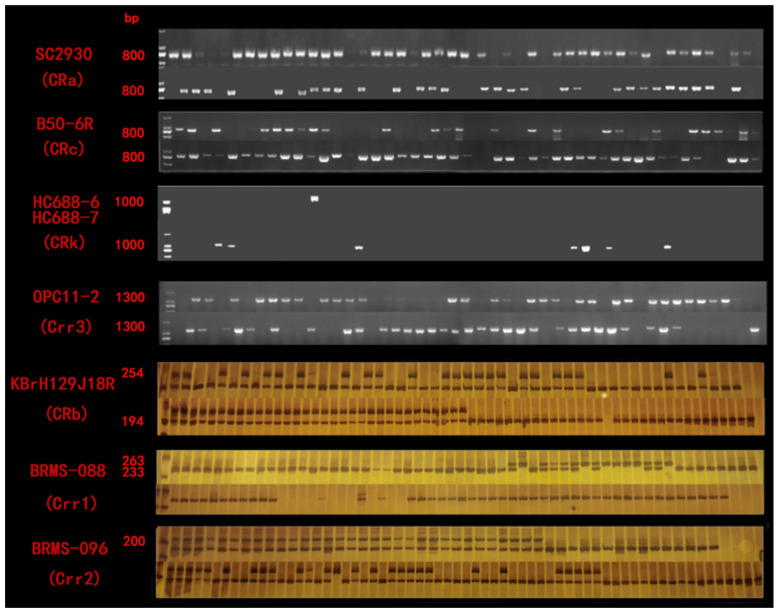
Amplification results of different disease resistance genes in the experimental materials.

**Figure 2 genes-13-01600-f002:**
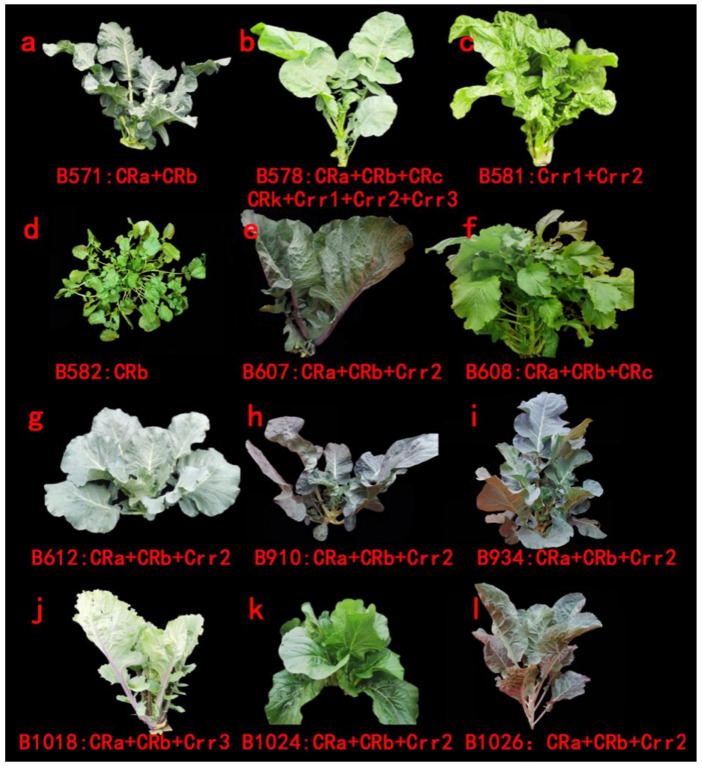
Species-aggregated multiple resistance sites. (**a**–**l**) letters represented experimental materials of B571, B578, B581, B582, B607, B608, B612, B910, B934, B1018, B1024, and B1026, respectively.

**Figure 3 genes-13-01600-f003:**
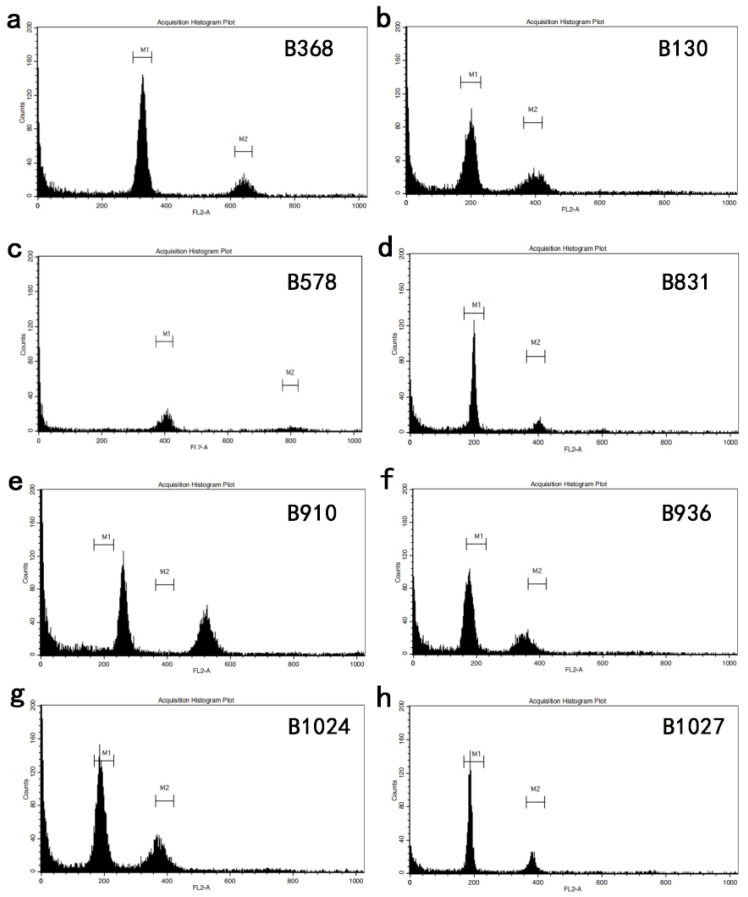
Detection of ploidy by flow cytometry. The letters (**a**–**h**) represented experimental materials of B368, B130, B578, B831, B910, B936, B1024, and B1027, respectively.

**Figure 4 genes-13-01600-f004:**
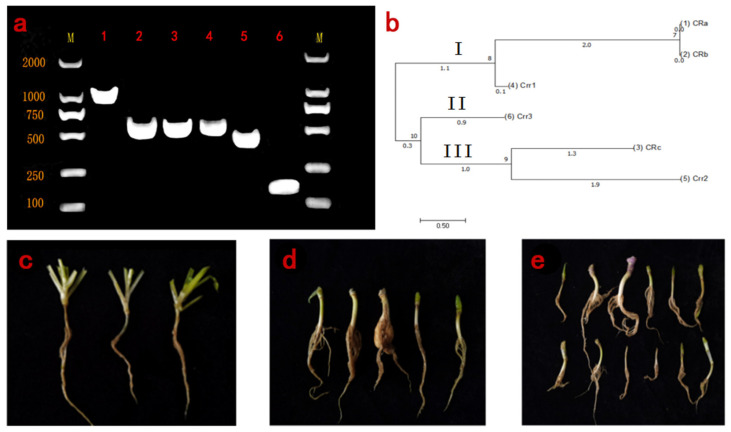
Identification of pathogen and disease test of the materials. Corresponding primers in (**a**) 1–6 represent PBRA_007750-2, Novel342-1, Novel407-2, PBRA_088439-1, PBRA_009348-1 and actin-1, (**b**) was the evolutionary tree of the CR genes, (**c**) was the in vitro identification results of “Yacui91” and (**d**,**e**) shows the in vitro identification results of hybrids from cabbage crossed with turnip.

**Figure 5 genes-13-01600-f005:**
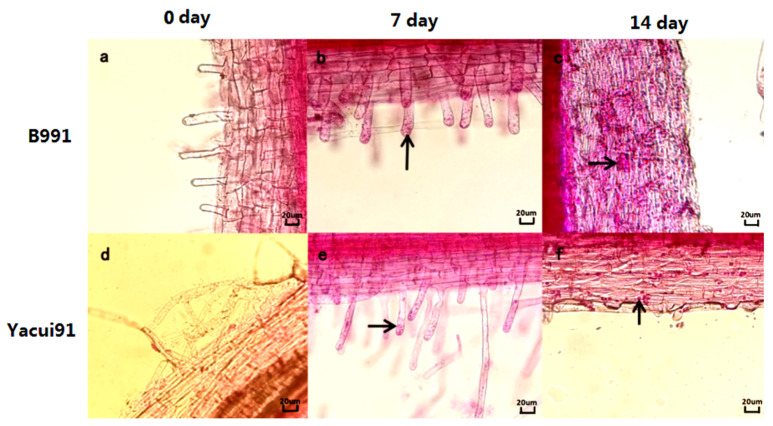
The process of *P. brassicae* infecting the root cells of resistant and susceptible broccoli varieties. The arrows indicated the conidia and sporangium. (**a**–**c**) present the profile of susceptible broccoli B991 on the 0th, 7th and 14th day of inoculation with *P. brassicae*. (**d**–**f**) present the profile of resistant broccoli “Yacui91” on the 0th, 7th and 14th day of inoculation with *P. brassicae*.

**Figure 6 genes-13-01600-f006:**
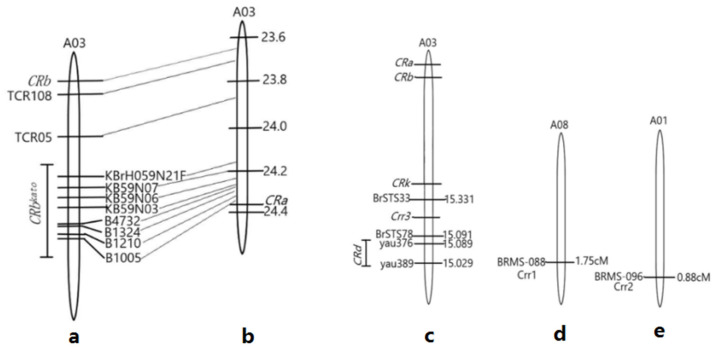
Identification of pathogen and disease test of the materials. Corresponding primers in (**a**) 1–6 represent PBRA_007750-2, Novel342-1, Novel407-2, PBRA_088439-1, PBRA_009348-1 and actin-1, (**b**) was the evolutionary tree of the CR genes, (**c**) was the in vitro identification results of “Yacui91”, (**d**,**e**) shows the in vitro identification results of hybrids from cabbage cross with turnip.

**Table 1 genes-13-01600-t001:** The genetic information of Information of 41 Cruciferae accessions.

Accessions	Species	Generations	Source Origin	Disease Resistant	Year
B891	Tuscan kale (*Brassica oleracea* L. var. *acephala*)	F_12_	CAAS-IVF, Beijing, China	S	2020
B991	Broccoli (*Brassica oleracea* L. var. *italica*)	F_6_	CAAS-IVF, Beijing, China	S	2020
B994	Broccoli (*Brassica oleracea* L. var. *italica*)	F_1_	Syngenta, China	S	2020
B1007	Chinese black moss (*Brassica campestris* L.var. *purpurea Baileysh*)	OP	Hubei, China	S	2020
B1081	Black mustard (*Brassica nigra*)	F_12_	CAAS-IVF, Beijing, China	S	2020
B1082	Abyssinian mustard (*Brassica carinata*)	F_12_	CAAS-IVF, Beijing, China	S	2020
B1083	Black mustard (*Brassica nigra*)	OP	CAAS-IVF, Beijing, China	S	2020
B1084	Black mustard (*Brassica nigra*)	OP	CAAS-IVF, Beijing, China	S	2020
B1086	Black mustard (*Brassica nigra*)	OP	CAAS-IVF, Beijing, China	S	2020
B359	Broccoli (*Brassica oleracea* L. var. *italica*)	BC1	CAAS-IVF, Beijing, China	S	2021
B366	Broccoli (*Brassica oleracea* L. var. *italica*) Wild cabbage (*Brassica macrocarpa* Guss.)	BC1	CAAS-IVF, Beijing, China	S, R	2021
B368	Broccoli (*Brassica oleracea* L. var. *italica*) Wild cabbage (*Brassica macrocarpa* Guss.) Rape (*Brassica napus* L.)	BC1	CAAS-IVF, Beijing, China	S, R, MR	2021
B369	Broccoli (*Brassica oleracea* L. var. *italica*) Wild cabbage (*Brassica macrocarpa* Guss.) Rape (*Brassica napus* L.)	BC1	CAAS-IVF, Beijing, China	S, R, MR	2021
B571	Broccoli (*Brassica oleracea* L. var. *italica*) Turnip (*Brassica rapa* L. ssp. *rapa*)	F_1_	CAAS-IVF, Beijing, China	HR	2021
B578	Rape (*Brassica napus* L.) (Oil rape)	F_1_	CAAS-IVF, Beijing, China	HR	2021
B581	Turnip (*Brassica rapa* L. ssp. *rapa*)	F_11_	CAAS-IVF, Beijing, China	HR	2021
B582	Yellow rocket (*Barbarea vulgaris* R. Br.)	F_11_	CAAS-IVF, Beijing, China	I	2021
B606	Red cabbage (*Brassica oleracea L.* var. *capitate*) Turnip (*Brassica rapa* L. ssp. *rapa*)	BC1F_3_	CAAS-IVF, Beijing, China	S, R, MR	2021
B607	Red cabbage (*Brassica oleracea L.* var. *capitate*) Turnip (*Brassica rapa* L. ssp. *rapa*)	BC1F_3_	CAAS-IVF, Beijing, China	S, R, MR	2021
B608	Black mustard (*Brassica nigra*)	BC1F_3_	CAAS-IVF, Beijing, China	S	2021
B611	Red cabbage (*Brassica oleracea L.* var. *capitate*) Turnip (*Brassica rapa* L. ssp. *rapa*)	BC1F_3_	CAAS-IVF, Beijing, China	S, R	2021
B612	Red cabbage (*Brassica oleracea L.* var. *capitate*) Turnip (*Brassica rapa* L. ssp. *rapa*)	BC1F_3_	CAAS-IVF, Beijing, China	S, R	2021
B613	Red cabbage (*Brassica oleracea L.* var. *capitate)* Turnip (*Brassica rapa* L. ssp. *rapa*)	BC1F_3_	CAAS-IVF, Beijing, China	S, R	2021
B614	Red cabbage (*Brassica oleracea L.* var. *capitate*) Turnip (*Brassica rapa* L. ssp. *rapa*)	BC1F_3_	CAAS-IVF, Beijing, China	S, R	2021
B621	Red cabbage (*Brassica oleracea L.* var. *capitate)* Kohlrabi (*Brassica oleracea L.* var. *caulorapa*)	BC2	CAAS-IVF, Beijing, China	S, R	2021
B831	Broccoli (*Brassica oleracea* L. var. *italica*) Wild cabbage (*Brassica macrocarpa* Guss.) Rape (*Brassica napus* L.)	BC1	CAAS-IVF, Beijing, China	S, R	2021
B832	Broccoli (*Brassica oleracea* L. var. *italica*) Wild cabbage (*Brassica macrocarpa* Guss.) Rape (*Brassica napus* L.)	BC1	CAAS-IVF, Beijing, China	S, R	2021
B908	Broccoli (*Brassica oleracea* L. var. *italica*) Turnip (*Brassica rapa* L. ssp. *rapa*) Rape (*Brassica napus* L.)	F_1_	CAAS-IVF, Beijing, China	S, R, HR	2021
B909	Broccoli (*Brassica oleracea* L. var. *italica*) Turnip (*Brassica rapa* L. ssp. *rapa*)	F_1_	CAAS-IVF, Beijing, China	S, R, HR	2021
B910	Broccoli (*Brassica oleracea* L. var. *italica*) Turnip (*Brassica rapa* L. ssp. *rapa*)	F_1_	CAAS-IVF, Beijing, China	S, R, HR	2021
B932	Broccoli (*Brassica oleracea* L. var. *italica*) Turnip (*Brassica rapa* L. ssp. *rapa*)	F_1_	CAAS-IVF, Beijing, China	S, R, HR	2021
B933	Broccoli (*Brassica oleracea* L. var. *italica)* Turnip (*Brassica rapa* L. ssp. *rapa*)	F_1_	CAAS-IVF, Beijing, China	S, R, HR	2021
B934	Broccoli (*Brassica oleracea* L. var. *italica*) Turnip (*Brassica rapa* L. ssp. *rapa*)	F_1_	CAAS-IVF, Beijing, China	S, R, HR	2021
B935	Broccoli (*Brassica oleracea* L. var. *italica)* Turnip (*Brassica rapa* L. ssp. *rapa*)	F_1_	CAAS-IVF, Beijing, China	S, R, HR	2021
B936	Broccoli (*Brassica oleracea* L. var. *italica*) Turnip (*Brassica rapa* L. ssp. *rapa*)	BC1	CAAS-IVF, Beijing, China	S, R, HR	2021
B1018	Broccoli (*Brassica oleracea* L. var. *italica*) Cabbage (*Brassica oleracea* L. var. *capitata*) Turnip (*Brassica rapa* L. ssp. *rapa*)	F_1_	CAAS-IVF, Beijing, China	S, R, HR	2021
B1019	Broccoli (*Brassica oleracea* L. var. *italica*) Cabbage (*Brassica oleracea* L. var. *capitata*) Turnip (*Brassica rapa* L. ssp. *rapa*)	F_1_	CAAS-IVF, Beijing, China	S, R, HR	2021
B1024	Choi Sum (*Brassica campestris* L. ssp. *chinensis* var. *utilis Tsen et Lee*) Chinese cabbage (*Brassica pekinensis Rupr.*)	F_1_	CAAS-IVF, Beijing, China	S, R, HR	2021
B1025	Choi Sum (*Brassica campestris* L. ssp. *chinensis* var. *utilis Tsen et Lee*) Chinese cabbage Chinese cabbage (B*rassica pekinensis Rupr.*) Turnip (*Brassica rapa* L. ssp. *rapa*)	F_1_	CAAS-IVF, Beijing, China	S, R, HR	2021
B1026	Broccoli (*Brassica oleracea* L. var. *italica*) Rape (*Brassica napus* L.)	F_1_	CAAS-IVF, Beijing, China	S, R, HR	2021
B1027	Broccoli (*Brassica oleracea* L. var. *italica*) Rape (*Brassica napus* L.)	BC1	CAAS-IVF, Beijing, China	S, R, HR	2021

**Table 2 genes-13-01600-t002:** Information of selected specific CR markers.

Primer Names	Loci	Primer Sequences (5′-3′)	Product Size (bp)
C2930-T-FW	*CRa*	TAGACCTTTTTTTTGTCTTTTTTTTTACCT	800
SC2930-R-FW	*CRa*	CAGACTAGACTTTTTGTCATTTAGACT	800
SC2930-RV	*CRa*	AAGGCCATAGAAATCAGGTC	800
KBrH129J18R-F	*CRb*	AGAGCAGAGTGAAACCAGAACT	254
KBrH129J18R-R	*CRb*	GTTTCAGTTCAGTCAGGTTTTTGCAG	194
B50-C9-FW	*CRc*	GATTCAATGCATTTCTCTCGAT	800
B50-6R-FW	*CRc*	AATGCATTTTCGCTCAACC	800
B50-RV	*CRc*	CGTATTATATCTCTTTCTCCATCCC	800
HC688-4-FW	*CRk*	TCTCTGTATTGCGTTGACTG	1000
HC688-6-RV	*CRk*	ATATGTTGAAGCCTATGTCT	1000
HC688-7-RV	*CRk*	AAATATATGTGAAGTCTTATGATC	1000
BRMS-088T	*Crr1*	TATCGGTACTGATTCGCTCTTCAAC	263
BRMS-088R	*Crr1*	ATCGGTTGTTATTTGAGAGCAGATT	233
BRMS-096T	*Crr2*	AGTCGAGATCTCGTTCGTGTCTCCC	220
BRMS-096R	*Crr2*	TGAAGAAGGATTGAAGCTGTTGTTG	189
OPC11-2ST	*Crr3*	GTAACTTGGTACAGAACAGCATAG	1300
OPC11-2SR	*Crr3*	ACTTGTCTAATGAATGATGATGG	1000

**Table 3 genes-13-01600-t003:** The information of clubroot pathogen identification markers.

Primer Names	Sequences (5′-3′)	Tm (°C)	Products (bp)
Actin1	F: GGGACATCACCGACTACCTG	57	160
	R: ACTGCTCCGAGTTGGACATC		
Novel342-2	F: CCACGCCTATACCCGGAAAG	58	666
	R: CAACAGGACGGCGTTGAAAG		
Novel407-2	F: GTCGTTGTTCGGGGAGAAGT	58	683
	R: GTCCATAGGTGTGGGAACGG		
PBRA_007750-2	F: ATCTGTTCGATTCGCCTGCT	58	1034
	R: GAGTGTACAGGCTCGCTCAG		
PBRA_008439-1	F: TCGGCGACCTGAGCGAGAA	58	651
	R: TCAACATGCGCATAGTAC		
PBRA_009348-1	F: CACTGCTATCGTCTCCCTGG	57	509
	R: CCTGCAATGTTTCGCTGCAA		

**Table 4 genes-13-01600-t004:** Identification of 41 cruciferous genotypes by CR markers.

Number	CRa	CRb	CRc	CRk	Crr1	Crr2	Crr3
B891	/	/	/	/	R	R	R
B991	/	/	/	/	R	R	R
B1007	/	/	/	/	R	/	R
B1081	/	/	/	/	R	R	R
B1082	/	/	/	/	R	R	R
B1083	/	/	/	/	/	/	R
B1084	/	/	/	/	/	/	R
B1086	R	/	/	/	R	/	R
B359	S	S	/	/	/	/	S
B366	S	S	/	/	/	S	S
B368	S	S	R	/	/	S	S
B369	S	S	R	/	/	S	S
B571	/	H	/	/	/	/	/
B578	H/S	H/R	H/S	R	H	H/R	S
B581	S	S	/	/	H	H	/
B582	/	R	/	/	/	/	/
B606	H/S	H	S	/	/	H/S	R
B607	H/S	H	S	/	/	H	S
B608	S	H	R	/	/	S	/
B611	H/S	H	S	/	/	S	S
B612	H/R/S	H	S	/	/	H/R/S	/
B613	H/S	H	R	/	S	H/S	S
B614	H/S	H/RS	R	/	S	H/R	S
B621	S	S	/	/	/	S	S
B831	S	S	/	/	S	S	S
B832	S	S	/	/	S	S	S
B908	H/S	H/RS/S	/	/	S	/	S
B909	H/R/S	H/R/S	/	/	S	/	S
B910	H/S	HS	/	/	S	S	R
B932	H/S	S	/	/	S	/	/
B933	S	S	/	/	/	S	R
B934	H/S	H/S	/	/	H/S	R/S	R
B935	H/R/S	H/S	/	/	H/S	H/R/S	R
B936	H/S	H/S	/	/	S	S	R
B1018	H/S	H/S	/	/	S	/	R
B1019	H/S	H/S	/	/	S	H	R
B1024	R/S	H/S	S	S	S	H	S
B1025	H/R/S	H/S	S	R/S	S	S	S
B1026	S	S	S	R	S	S	R/S
B1027	S	S	/	/	S	S	R

Note: R indicates a homozygous disease resistance site, S indicates a homozygous susceptibility site and H indicates a heterozygous disease resistance site and/or an undetected disease resistance site.

## Data Availability

The data presented in this study are available upon request from the corresponding author.

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
