# Peer review of "Germplasm Enhancement and Identification of Loci Conferring Resistance against Plasmodiophora brassicae in Broccoli"

_genes, 2022, doi:10.3390/genes13091600_

Round 1
Reviewer 1 Report
The manuscript on Germplasm enhancement and identification of loci conferring 2 resistance against Plasmodiophora brassicae in broccoli is a good study and an apt for the brassica researchers. The new cultivar or genotype is very much needed with better resistance to Club Root.
Author Response
Dear Professor,
Thanks for your good comments.
Sincerely yours,
Dr. Zhansheng Li
Reviewer 2 Report
In the current manuscript, Xie et al. tried to screen clubroot-resistant germplasms from cruciferous materials collected in 2020 and 2021 based on the reported P. brassicae-resistant materials (turnip, cabbage, oilseed rape) and identify molecular markers to rapidly detect clubroot resistance among a variety of resources, and this would facilitate resource innovation and improvement of materials in the Brassica genus to develop varieties with multiple resistance sites and identify the associated disease resistance genes. All parts of the manuscript are interesting and clearly summarize data valuable for the research community.
-TITLE
The paper title is well stated, it is informative and concise.
-ABSTRACT,
Abstract needs improvement.
-INTRODUCTION
The introduction was not well written, and it is too briefly presenting the subject and research problem. There is a lot of data in the literature- The introduction should be improved
-MATERIAL AND METHODS
Material and research methods are presented appropriately. The experimental setup and the description in the methods section are well structured, and the statistical analysis is done alright.
-RESULTS
The results obtained in this study are interesting. Results presented correctly.
-DISCUSSION
In general, the discussion of the results and conclusions is correct, but not sufficient. The topic was not well discussed. The authors do not make full use of the literature resources.
-CONCLUSIONS
Repeated abstract of the work. This part needs to be improved.
Author Response
Dear Professor,
I am very grateful to your good comments for the manuscript, and we amended the relevant part in manuscript. Some of your questions were answered below.
1. -INTRODUCTION
The introduction was not well written, and it is too briefly presenting the subject and research problem. There is a lot of data in the literature- The introduction should be improved.
Response: Yes, I really agree with you and we had improved this part, please check it.
2. -DISCUSSION
In general, the discussion of the results and conclusions is correct, but not sufficient. The topic was not well discussed. The authors do not make full use of the literature resources.
Response: Thanks for your good comments and advices, we have modified some statements and added new discussion, please check it.
3. -CONCLUSIONS
Repeated abstract of the work. This part needs to be improved.
Response: I am sorry for this mistake, we have revised this part, please check it.

Reviewer 3 Report
The authors work on identifying loci conferring resistance against Plasmodiophora brassicae in broccoli. I have some comments and suggestions that must be addressed before acceptance in the journal. The general and specific comments are suggested to be taken into consideration by the authors below:
General comments
-The article has a great novelty to publish in this journal, “Genes”.
-The manuscript presents interesting data with innovative findings.
- The manuscript falls into the scope of the journal.
-In the abstract, the authors should include some sentences highlighting the background statement and objective of the study. Add conclusive remarks and/or recommendations at the end of the abstract.
-The introduction needs to be improved with proper background, justification, and hypotheses of the study.
-The methodology is sound enough to interpret the data and draw a conclusion.
-There are a few typos, errors and grammatical mistakes that must be addressed.
The discussion needs to improve by interpreting and explaining their results by citing relevant references.
-Add a conclusion and write major findings of the study with conclusive remarks and recommendations.
Specific comment
1. Abstract
the abstract is disjointed, and I found it lacks some basic layout here. I highlighted the sequence to follow in improving the abstract:
- The overall purpose of the study and the research problem
- Materials and methods;
- Major findings as a result of your analysis;
- a summary of your interpretations;
- Conclusions.
2. Introduction
The introduction section could be expanded by results relating to the subject matter. I found the paragraph arrangement to be disjointed. Here I recommend that paragraph two be move to paragraph one and vice versa.
Line 32: too many references
Line 42-44: source/reference is lacking
Line 49: too many references
3. Materials and methods
- What is the choice of selecting F1 and BC1 considering that these lines are still subjected to segregation
-Line 86 to 88: I found this statement inconclusive as the final PCR reaction contained 10.0 μL reaction, whereas going by the information provided, “2.0 μL of DNA template, 2.0 μL of buffer, 0.5 μL of primer, and 5.0 μL of Taq Plus Master Mix” makes 9.5 μL.
4. Conclusion
-line 291: change “wo” to “we”
Add a section of the conclusion. Write major findings of the study with conclusive remarks and recommendations.
Author Response
Dear Professor,
I am very grateful to your good comments for the manuscript, and we amended the relevant part in manuscript. Some of your questions were answered below.
1. Abstract, the abstract is disjointed, and I found it lacks some basic layout here. I highlighted the sequence to follow in improving the abstract:
- The overall purpose of the study and the research problem
- Materials and methods;
- Major findings as a result of your analysis;
- a summary of your interpretations;
- Conclusions.
Response:
Yes, I really agree with you and thanks for your good comments and suggestions. In the revised manuscript, we had improved this part, please check it.
2. Introduction
The introduction section could be expanded by results relating to the subject matter. I found the paragraph arrangement to be disjointed. Here I recommend that paragraph two be move to paragraph one and vice versa.
Line 32: too many references
Line 42-44: source/reference is lacking
Line 49: too many references.
Response:
Thanks for your good comments and serious review, we have modified this part and improved references in all suggested lines, please check it.
3. Materials and methods
- What is the choice of selecting F1 and BC1 considering that these lines are still subjected to segregation
-Line 86 to 88: I found this statement inconclusive as the final PCR reaction contained 10.0 μL reaction, whereas going by the information provided, “2.0 μL of DNA template, 2.0 μL of buffer, 0.5 μL of primer, and 5.0 μL of Taq Plus Master Mix” makes 9.5 μL.
Response:
Yes, you are right, we just provided meaningful work and new cruciferous species useful for breeding of broccoli and the other Brassica. Although these F1 and BC1 should be still purified by self or crossing selecting with better background, we firstly provide these medium materials beneficial to deep researches in Brassica crops and international corporations in exchanging materials better for global breeding in broccoli, Chinese cabbages, rapeseed and so on. At same time, I am very sorry for this mistake Line 86 to 88, there are one pair of primers, so “0.5 μL of primer” should be modified as “1.0 of primers”, we have modified it, please check it.
4. Conclusion
-line 291: change “wo” to “we”
Add a section of the conclusion. Write major findings of the study with conclusive remarks and recommendations.
Response:
For line 291, I am sorry and we have changed this mistake. And meanwhile, conclusions had been revised and improved with help of your advices, thank you very much.
Yours Sincerely,
Dr. Zhansheng Li

Round 2
Reviewer 2 Report
All my comments have been addressed. I think that the present form of the manuscript can be published in Genes.